# Path-LLM: A Multi-Modal Path Representation Learning by Aligning and Fusing with Large Language Models

## ABSTRACT

The advancement of intelligent transportation systems has led to a growing demand for accurate path representations, which are essential for tasks such as travel time estimation, path ranking, and trajectory analysis. However, traditional path representation learning (PRL) methods often focus solely on single-modal road network data, overlooking important physical and regional factors that influence real-world traffic dynamics. To overcome this limitation, we introduce **Path-LLM**, a multi-modal path representation learning model that integrates large language models (LLMs) into PRL. Our approach leverages LLMs to interpret both topological and textual data, enabling robust multi-modal path representations. To effectively align and merge these modalities, we propose *TPalign*, a contrastive learning-based pretraining strategy that ensures alignment within the embedding space. We then present *TPfusion*, a multimodal fusion module that dynamically adjusts the weight of each modality before integration. To further optimize LLM training, we introduce a *Two-stage Overlapping Curriculum Learning (TOCL)* approach, which progressively increases the complexity of the training data. Finally, we evaluate Path-LLM on two real-world datasets across traditional PRL downstream tasks, achieving up to a 61.84% improvement in path ranking performance on the Xi'an dataset. Additionally, Path-LLM demonstrates superior performance in both few-shot and zero-shot learning scenarios. Our code is available at: https://anonymous.4open.science/r/Path-LLM-F053.

## KEYWORDS

Path representation learning, Large language models, Curriculum learning, Contrastive Learning

## 1 INTRODUCTION

Paths are fundamental to numerous real-world intelligent transportation applications and mapping services, such as travel time estimation [8, 16, 21, 35], path ranking [34, 36, 37], and trajectory analysis [15, 26, 27, 29]. As web-based transportation systems expand in both scale and complexity, the demand for accurate path representations has surged, positioning path representation learning (PRL) a key research focus [11, 38]. PRL aims to generate generalizable representations of paths, enabling their effective use across a diverse range of downstream tasks.

Previous PRL methods [32, 33, 35, 36] predominantly rely on single-modal data, typically focusing on road network graphs that emphasize the connectivity and spatial relationships between road segments. However, a high-quality path representation should incorporate multiple factors, including physical (e.g., road length and width) and functional (e.g., road name and type) characteristics, which capture the essential attributes of paths and are crucial for downstream tasks. For example, as illustrated in Figure 1, there are two potential routes from point $S$ to $D$: $P_1 = \langle e_1, e_2, e_3, e_4, e_5 \rangle$ and $P_2 = \langle e_7, e_8, e_6 \rangle$. While both paths cover similar distances, their differing physical and functional features affect travel efficiency.

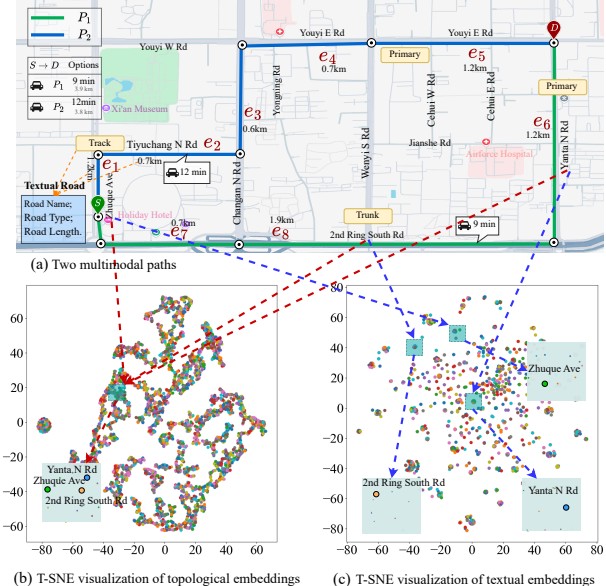

(a) Two multimodal paths

(b) T-SNE visualization of topological embeddings

(c) T-SNE visualization of textual embeddings

**Figure 1: An Example of Two Paths with Multi-modality Information. Road segments are described using names, lengths, and types. Road names are often named based on their geographic location, local landmarks, or road type, offering insights into continuity and significance and function. Length helps assess each segment's importance, while type informs the model about its structural and functional role.**

$P_1$, which traverses *'2nd Ring South Rd'*—a major trunk road with smooth traffic flow and fewer traffic lights—takes just 3 minutes to travel. In contrast, $P_2$, which follows *'Tiyuchang N Rd'* and experiences more interruptions and congestion, takes 10 minutes. Therefore, it is essential to develop a path representation learning method that integrates multimodal information to accurately capture path dynamics.

Recently, large language models (LLMs) have achieved remarkable success in natural language processing (NLP) [2, 6, 7, 19, 30], excelling in both textual and sequential modeling. Given that the physical, regional, and functional information of paths can be expressed in textual form, and that paths can be represented as sequences of road segments—analogous to sentences—LLMs hold significant potential for advancing path representation learning. LLMs' advanced NLP capabilities allow them to effectively handle the textual information within paths. Additionally, their strong performance in few-shot and zero-shot tasks demonstrates their ability to learn path representations with high generalization, even in data-scarce scenarios such as regions with limited road data.

However, LLMs struggle to model the complex topological structures inherent in paths, posing two major challenges for PRL, which are outlined as follows.

(1) **How can the topological and textual modalities be effectively integrated to ensure the model captures both the spatial structure and semantic features of the paths?** Due to the differences between these modalities, two consecutive road segments in a path might be spatially close in the topological embedding space but distant in the textual embedding space. For instance, in Figure 1 (a), $e_8$ and $e_6$ denote edges on *'2nd Ring South Rd East Section'* and *'Yanta N Rd'*, respectively, which are close in the topological embedding space (cf. Figure 1 (b)), yet their textual embeddings are far apart (cf. Figure 1 (c)). Thus, it is crucial to develop an effective fusion mechanism that ensures the complementarity of topological and textual information.

(2) **How can LLMs be trained to maintain strong generalization capabilities despite discrepancies in training difficulties?** Continuing with the previous point, the degree of discrepancy between the topological and textual embeddings of different segments can vary considerably. For example, $e_1$, $e_8$, and $e_6$ are all close in the topological embedding space (cf. Figure 1 (b)), but the distance between the textual embeddings of $e_6$ and $e_8$ is larger compared to that between $e_1$ and $e_8$ (cf. Figure 1 (c)), making it more challenging for the model to align topology and textual embeddings. Therefore, the training process must account for these variations in alignment difficulty to ensure effective fusion and improve the model's generalization performance.

To address these challenges, we propose **Path-LLM**, a multimodal path representation learning model that effectively leverages the power of LLMs to integrate both topological and textual modalities. To address the first challenge, we propose a contrastive learning-based pretraining strategy, namely *TPalign*, and a multimodal fusion module, namely *TPfusion*. To mitigate conflicts between modalities, *TPalign* is first used to pretrain the path encoder, aligning topological and textual modalities within the embedding space, which lays the groundwork for effective multimodal fusion. To further enhance information complementarity, we design the *TPfusion* module, which employs an adaptive gating mechanism to dynamically adjust the weight distribution between modalities, enabling a more refined integration of topological and textual embeddings with precise importance balancing. To address the second challenge, we propose the **T**wo-stage **O**verlapping **C**urriculum **L**earning method (*TOCL*), which incorporates two key stages: 1) *single-step overlapping training* and 2) *two-stage curriculum learning*. The former ensures a smoother transition from easier to more difficult samples, while the latter helps to prevent catastrophic forgetting and enhances the model's generalization capabilities. These two components are particularly crucial in multimodal training, where alignment discrepancies between modalities can vary significantly. Further details will be discussed in Section 4. Our main contributions are summarized as follows:

- We propose a multimodal path representation learning model Path-LLM, which uses a partially frozen LLM to interpret the topological, physical, regional, and functional information of paths. To the best of our knowledge, this is the first work that introduces LLMs into path representation learning.

- To facilitate multimodal integration, we propose multimodal align and fusion modules, i.e., TPalign and TPfusion, respectively. Specifically, TPalgin aims to align the textual and topological embeddings, and then TPfusion can model the interactions between modalities.

- To improve the model's generalization capability, we propose a two-stage overlapping curriculum learning method to prevent catastrophic forgetting.

- We conduct extensive experiments on two real-world datasets, Xi'an and Chengdu, across two downstream tasks to evaluate the performance of Path-LLM. The results demonstrate that Path-LLM consistently outperforms the baselines in all tasks, showing superior generalization and fusion capabilities.

## 2 RELATED WORK

### 2.1 Path Representation Learning

Path representation learning (PRL) is an important research field for understanding and predicting patterns in transportation networks. Its main goal is to learn universal low-dimensional embeddings that capture both road segment attributes and topological information. Compared to the specific tasks-oriented methods [5, 8, 20, 21, 31, 37], PRL can be applied to a variety of downstream tasks such as travel time estimation and path ranking. Recently, several in-depth studies [17, 32, 33, 35] have been conducted on PRL. PIM [32] is an unsupervised learning framework that relies on high-quality negative samples for training path encoders through mutual information maximization. However, generating effective negative samples in data-scarce environments remains a challenge. WSCCL [33] introduces curriculum learning strategies but suffers from high model complexity and computational cost, limiting its application to large-scale datasets. LightPath [35] relies on sparse autoencoders and global local knowledge distillation methods. In comparison, our method simplifies data preparation and training processes. JGRM [24] presents a trajectory representation learning framework based on GPS trajectory modality and topological route modality based on MLM Loss and Match Loss, respectively. Although methods such as PIM, WSCCL, and Light-Path focus solely on the topological modality, and while JGRM incorporates both GPS trajectory and topological modalities, there is a noticeable gap in the exploration of path representations that consider both topological and textual modalities. In response to this, we propose **Path-LLM**, a novel approach that integrates both topological and textual modalities to learn a more comprehensive and generic path representation.

### 2.2 Cross Domain Application of LLM

Recent advances in large language models (LLMs) have demonstrated their potential across various multimodal applications, including time series analysis [18, 28, 39] and computer vision tasks [1, 3, 12, 14]. These approaches typically employ domain-specific encoders to map data into representations for LLMs, facilitating downstream tasks. These methods commonly use domain-specific encoders to convert sample data into embeddings for LLMs, aiding downstream tasks. While the mentioned studies offer valuable insights, their methods are specific to their fields, and cannot be directly applied to PRL. To our knowledge, no existing research

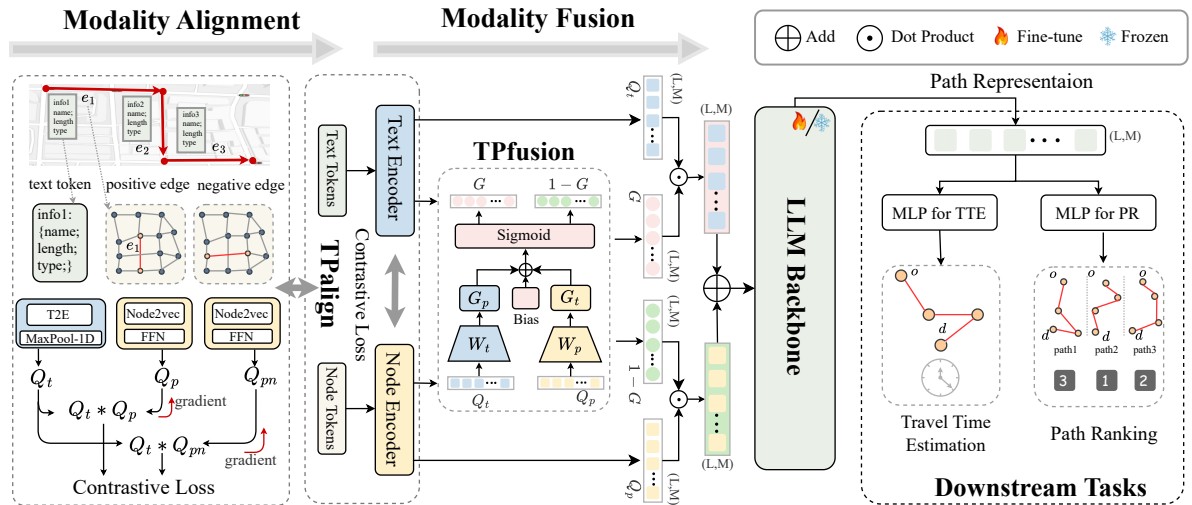

**Figure 2: Path-LLM Overview**

systematically focuses on applying LLMs to path representation learning.

## 3 PRELIMINARY

In this section, we introduce the basic definitions and concepts required to understand our proposed path representation learning framework, including definitions related to road networks, multimodal paths, and path representations.

### 3.1 Basic Definitions

DEFINITION 1. **Road Network.** *A road network is represented as a graph $\mathcal{G} = (V, E)$, where $V$ is the set of nodes $v_i$ representing road intersections, and $E \subseteq V \times V$ is an edge set, where $e_i = (v_j, v_k)$ represents a road segment from $v_i$ to $v_j$.*

DEFINITION 2. **Topological Path.** *A topological path is defined as $p = \langle e_1, e_2, \cdots, e_L \rangle$, which consists of an ordered sequence of edges, where two consecutive edges $e_i = (v_i, v_k)$ and $e_{i+1} = (v_k, v_{i+1}) \in E$ share a common vertex, denoted as $e_i \cap e_{i+1} = v_k$, and $L$ is the length of path $p$.*

DEFINITION 3. **Textual Path.** *Given a topological path $p$, a textual path is defined as $t = \langle a_1, a_2, \cdots, a_L \rangle$, which $a_i$ denotes the textual description (e.g., road name, type and length, etc.) of edge $e_i$.*

DEFINITION 4. **Edge Representation.** *In a road network graph, the topological representation of an edge is denoted by the embedding $h_p \in \mathbb{R}^{D_p}$, which is obtained from graph embedding models such as Node2Vec [10], and the textual representation of an edge is denoted by $h_t \in \mathbb{R}^{D_t}$ that is obtained by text-embeddings-3 [1] from OpenAI.*

---

[1] https://platform.openai.com/docs/guides/embeddings

## 3.2 Problem Definitions

In a graph network $\mathcal{G}$, the objective of multimodal path representation learning for a path set $\mathbb{P} = \{p_i, t_i\}_{i=1}^{N}$ is to learn a function $F_\theta(\cdot)$ that can generate a generic multi-modal path representation $O \in \mathbb{R}^M$ for each path $p \in \mathbb{P}$, which can be used for the downstream tasks. It can be expressed as follows:

$$O = F_\theta(p, t) : (\mathbb{R}^{L \times D_p}, \mathbb{R}^{L \times D_t}) \rightarrow \mathbb{R}^{L \times M}, \quad (1)$$

where $O$ is the learned generic multi-modal path representation, $\theta$ denotes the learnable parameters for the whole model, $L$ is the length of the input path, and $M$ is the total number of paths used in the training process. $D_p$ and $D_t$ denote the feature dimensions for the edge embeddings for topological and textual path, respectively, and $M$ is the output dimension for path representation.

## 4 METHODS

This section provides an overview of Path-LLM, which consists of four key components: a pre-trained modality align module (TPalign), a multimodal fusion module (TPfusion), a frozen pre-trained LLM, and a two-stage overlapping curriculum learning model (TOCL), as illustrated in Figures 2 and 4, respectively. The subsequent sections offer detailed descriptions of each component.

### 4.1 Input Embeddings

Road segments can be characterized by both spatial relationships and textual information. This duality offers complementary insights: topological embeddings elucidate the structural configuration of the road network, while textual embeddings facilitate the interpretation of human-readable information, such as the functions and attributes of specific road segments. The following section will specifically introduce the construction methods for the two types of embeddings.

**Figure 3: Instance-level CL and Feature-level CL.**

*4.1.1 Textual Path Construction.* To capture the key attributes, such as name, type, and length—for each segment in path $p$, we introduce LLMs to interpret the textual path. In particular, segment names offer insights into regional road characteristics; length information allows the model to assess the significance of each segment within the overall path. Additionally, road type information aids the model in understanding the functional and structural properties of various road segments, thereby enhancing path representations. We retrieve road segment names, lengths, and types from OpenStreetMap [13]. For a given topological path $p = \langle e_1, e_2, \cdots, e_L \rangle$, we can obtain the corresponding textual path $t = \langle a_1, a_2, \cdots, a_L \rangle$, where $a_i$ represents the textual information of $e_i$.

*4.1.2 Modality Embeddings.* As shown in the left-most part of Figure 2, we employ OpenAI's text-embeddings-3 (T2E) followed by a 1D Max-Pooling layer as the text encoder, while the edge encoder combines Node2Vec [10] with a Feed Forward Network (FFN). We use edge encoder and text encoder to obtain topological embeddings $Q_p$ and textual embeddings $Q_t$ for all road segments in a path, respectively, as defined in Equations 2 and 3.

$$H_p = Node2vec(p); \quad Q_p = FFN(H_p); \quad (2)$$

$$H_t = T2E(t); \quad Q_t = MaxPool1D(H_t), \quad (3)$$

where, $H_p \in \mathbb{R}^{L \times D_p}$, $H_t \in \mathbb{R}^{L \times D_t}$, $D_p$ and $D_t$ correspond to the feature dimensions of the path and text respectively. $Q_p \in \mathbb{R}^{L \times M}$, $Q_t \in \mathbb{R}^{L \times M}$, $M$ represents the input dimension of LLM.

## 4.2 Align and Fuse

The spatial nature of topological information in path limits LLMs' interpretation compared to textual data. To address it, we propose a multimodal fusion module, TPfusion, which aligns these modalities to enhance their interactions. Specifically, we introduce a contrastive learning-based pretraining module, TPalign, to align topological data with textual information. This alignment is essential for enabling LLMs to interpret path data more meaningfully and spatially awarely.

*4.2.1 Topological and Textual Path Modality Alignment (TPalign).* TPalign performs contrastive learning (CL) tasks at both the instance and feature levels to pretrain the Feed Forward Network (FFN)

of the edge encoder, as shown in Figure 3. At the instance level, it aligns the textual and topological embeddings of individual road segments, which focuses on minimizing the distance between textual embeddings and their corresponding topological embeddings for the same path, while maximizing the difference between embeddings of different paths. This design ensures that LLMs can better understand and utilize the spatial context in path representations. At the feature level, it aligns the embeddings across feature dimensions to facilitate TPfusion better integrating multimodal information across different feature dimensions.

Specifically, given a road segment $e_i \in E$, we denote the positive embeddings as $h_t^i$ and $h_p^i$, and negative embeddings as $h_t^i$ and $h_p^j$, where $i \neq j$. Our goal is to bring the positive pair closer in the embedding space and push the negative pair further apart.

In *instance-level CL*, we construct positive/negative pairs as $(q_t^i, q_p^i)$ and $(q_t^i, q_p^j)$, where $q_p^i = FFN(h_p^i)$ and $q_t^i = Maxpool1D(h_t^i)$. Then, the similarity of one pair embedding can be calculated as follows.

$$S(q_p^m, q_t^n) = \text{Sigmoid}(\text{MLP}(q_p^m || q_t^n)), \quad (4)$$

where $S(q_p^m, q_t^n)$ denotes the similarity function that measures the similarity of a pair of topological and textual path embeddings of two edges, $e_m$ and $e_n$, respectively. $\cdot || \cdot$ denotes the concatenation operation, and Sigmoid($\cdot$) denotes the Sigmoid function.

Let $y$ represent the label of pair embeddings, where positive pairs are assigned a value of 1 and negative pairs are assigned a value of 0. Then, we construct a set of training data as $\mathbb{B} = \{(\hat{q}_p^i, \hat{q}_t^i, y^i)\}$. Then, we calculate the instance-level contrastive loss as follows:

$$\mathcal{L}_{\text{ins}} = -\frac{1}{|\mathbb{B}|} \sum_{i=1}^{|\mathbb{B}|} [y^i \log S(\hat{q}_p^i, \hat{q}_t^i) + (1 - y^i) \log \log S(\hat{q}_p^i, \hat{q}_t^i)]. \quad (5)$$

In *feature-level CL*, we treat the textual embeddings corresponding to all road segments in each batch as a textual feature matrix $Q_t \in \mathbb{R}^{B \times M}$. Similar to the instance-level, we can obtain a positive feature matrix $Q_p^+$ and the negative $Q_p^-$, where $Q_p = [q_p^i]_{i=1}^B \in \mathbb{R}^{B \times M}$. The column embeddings of matrices $Q_t$ and $Q_p$ are represented as $c_t$ and $c_p$ respectively. For the $j$-th pair $(c_t^j, c_p^j)$ within $M$ columns, we compute the similarity as:

$$H(c_p^m, c_t^n) = \text{Sigmoid}(\text{MLP}(c_p^m || c_t^n)), \quad (6)$$

where $H(c_p^m, c_t^n)$ denotes the similarity function that measures the similarity of a pair of topological and textual path embeddings of two columns, $c_m$ and $c_n$, respectively.

Let $z$ represent the label of pair embeddings, where the positive column pair is assigned a value of 1 and the negative pair is assigned a value of 0. Then, we construct a set of training data as $\mathbb{M} = \{(\hat{c}_p^i, \hat{c}_t^i, z^i)\}$. The feature-level contrastive loss is computed as:

$$\mathcal{L}_{\text{fea}} = -\frac{1}{|\mathbb{M}|} \sum_{i=1}^{|\mathbb{M}|} [z^i \log H(\hat{c}_p^i, \hat{c}_t^i) + (1 - z^i) \log \log H(\hat{c}_p^i, \hat{c}_t^i)]. \quad (7)$$

Finally, the overall training loss is computed as a weighted combination of the instance-level and feature-level losses:

$$\mathcal{L} = \lambda \mathcal{L}_{\text{ins}} + (1 - \lambda) \mathcal{L}_{\text{fea}}, \quad (8)$$

where $\lambda$ is a hyperparameter that balances the contributions of the instance-level and feature-level contrastive losses.

*4.2.2 Topological and Textual Path Modality Fusion (TPfusion).* To improve the effective parsing of multimodal information by LLMs, it is necessary to fuse these two features. However, directly adding the textual and topological features or applying static weights may lead to conflicts between the modalities. To address this issue, we propose a multimodal fusion module, TPfusion, which dynamically adjusts the contribution of each modality by selectively weighting different positions. Specifically, the topological and textual embeddings $Q_p$ and $Q_t$ are fed into a gated fusion module to compute a gate value $G \in \mathbb{R}^{L \times M}$ based on the linear transformations of both input features, followed by a sigmoid activation function:

$$G = \text{Sigmoid}(W_p Q_p + W_t Q_t + b), \qquad (9)$$

where $W_p \in \mathbb{R}^{L \times M}$ and $W_t \in \mathbb{R}^{L \times M}$ are learnable weight matrices, $b \in \mathbb{R}^M$ is the bias term. The sigmoid function ensures that the gate values range between 0 and 1. The final fused representation $Q_f$ is obtained by combining $Q_p$ and $Q_t$ using the computed gate values, where the gate values control the contribution of each modality:

$$Q_f = G \odot Q_p + (1 - G) \odot Q_t, \qquad (10)$$

where $\odot$ denotes the element-wise multiplication operator.

### 4.3 Pre-Trained and Frozen LLM Block.

We employ a pre-trained GPT-2 model [19] to enhance path representation learning, capitalizing on its efficacy in sequential data processing. To maximize its embedded knowledge while minimizing computational costs, we utilize a selective fine-tuning strategy, as inspired by [39].

*4.3.1 Freezing Layers:* During the training phase, we freeze both the multi-head self-attention (MHA) and feed-forward network (FFN) layers in GPT-2. These components have already been trained to capture general patterns from large-scale language data, and freezing them prevents overfitting to the current task, while also reducing the training time and computational complexity. By freezing these layers, we retain the robust representational power of the pre-trained GPT-2 model.

*4.3.2 Fine-tuning Positional Embeddings and Layer Normalization:* To enable the model to adapt to path representation learning, we fine-tune specific layers that can better capture task-specific information. First, we fine-tune the word positional embeddings (WPE), allowing the model to better encode the sequential nature of paths. Paths inherently exhibit spatial and topological structures, which positional embeddings can effectively represent by learning the relative positions of road segments. Furthermore, we fine-tune the layer normalization (LN) layers to stabilize training and enhance model performance, ensuring that gradient updates remain stable during transformations.

*4.3.3 Skipping Input Embeddings:* We bypass the original input embedding layer in GPT-2 and use the fused representation $Q_f$ from TPfusion as the direct input, then we obtain a universal path representation $O \in \mathbb{R}^{L \times M}$.

### 4.4 Two-stage OverLapping Curriculum Learning

Fine-tuning large language models (LLMs) for specific tasks, particularly when integrating multiple modalities, poses a significant challenge in maintaining generalization capabilities. Traditional fine-tuning approaches often lead to overfitting on task-specific data, reducing the model's ability to generalize across diverse scenarios. Additionally, when handling complex multimodal data, the model may struggle to effectively balance the learning of both simple and complex examples, further impairing performance. To address these issues, we propose Two-stage OverLapping Curriculum Learning, termed as TOCL. The overview of TOCL is illustrated in Figure 4.

*4.4.1 Single Step Overlapping Training:* In this phase, TOCL adopts a progressive learning strategy, where the model $\theta_i$ is trained incrementally over $N$ rounds, with each round introducing increasingly complex data. Unlike traditional methods, TOCL selects the training data for each round using a sliding window approach, ensuring that each round's data partially overlaps with the previous round's data. This overlap allows the model to reinforce previously acquired knowledge while progressively integrating more challenging data.

The training data is ranked by Mean Squared Error (MSE), sorting paths from the easiest to the most challenging. For each round $i$, a window of size $\gamma_i$ is defined, with start and end indices denoted by $\mathbf{x}_i^{st}$ and $\mathbf{x}_i^{end}$ respectively. This approach ensures that the model encounters only simpler data in the early stages and progressively introduces more complex data as training advances. The data from round $i$ partially overlaps with the data from round $i-1$. This design aims to allow the model to review previously learned knowledge in each training round, thereby ensuring a smooth transition to learning more complex data.

*4.4.2 Two-Stage Curriculum:* TOCL further introduces a two-stage training strategy to balance both complex and fundamental knowledge retention. As illustrated in Figure 4, this process is divided into two phases: (1) Stage 1 (Round 1 to $K$): In the first $K$ rounds, the model is progressively trained on all data using the single-step overlapping approach. This phase allows the model to build a strong foundation by gradually increasing the complexity of the training data; (2) Stage 2 (Rounds $K + 1$ to $N$): From round $K + 1$ to round $N$, the model is trained on all data again. This ensures that the model not only retains and understands complex knowledge but also consolidates basic knowledge.

## 5 EXPERIMENTS

### 5.1 Datasets and Settings

We conduct experiments on real-world datasets from the cities of Chengdu and Xi'an. The original datasets, released by Didi, consist of GPS trajectories recorded by taxis in the two cities. We obtain the urban road networks from OpenStreetMap to perform map-matching on the trajectories. We remove paths shorter than 10 segments. Ultimately, the Chengdu dataset contains 4,315 road segments and 121,526 paths, while the Xi'an dataset includes 3,392 road segments and 94,917 paths.

Figure 4: The Training Process of TOCL

We divide each city's dataset into training, testing, and validation sets with a ratio of 8:1:1. We use GPT-2 as the base LLM for development, and all models are implemented using PyTorch [25]. We set the training process to 40 epochs. All experiments are conducted on Linux server with an NVIDIA RTX3090 24GB GPU. Our code is available at: https://anonymous.4open.science/r/Path-LLM-F053.

## 5.2 Downstream Tasks

In our study, we conduct experimental analysis on two downstream tasks:

**Travel Time Estimation (TTE):** We extract the travel time for each path from the original trajectory data. We make predictions using a fully connected layer, as described below:

$$\hat{y}_{\text{TTE}} = \text{FC}_{\text{TTE}}\left(O\right). \tag{11}$$

To evaluate the performance of TTE, we use Mean Absolute Error (MAE) [33], Mean Absolute Percentage Error (MAPE) [33] and Mean Absolute Relative Error (MARE) [33]. Lower values of these metrics indicate better performance.

**Path Ranking:** We consider the path used by drivers in historical trajectories as the optimal path. Then, for each optimal path, we generate multiple candidate paths connecting the same origin and destination using path finding algorithm [23]. Finally, we calculate the Jaccard similarity coefficient between the optimal path and each candidate path to determine the ranking score:

$$J(A, B) = \frac{|A \cap B|}{|A \cup B|}, \tag{12}$$

where $A$ and $B$ represent two sets, while $|A \cap B|$ and $|A \cup B|$ denotes the size of their intersection and union, respectively. Similar to TTE, we use a fully connected layer to predict the ranking score. In addition to MAE, we also use the Kendall rank correlation coefficient ($\tau$), and Spearman's rank correlation coefficient ($\rho$) as metrics for path ranking evaluation:

$$\tau = \frac{N_{\text{con}} - N_{\text{dis}}}{n(n-1)/2}, \tag{13}$$

$$\rho = 1 - \frac{6\sum_{i=1}^{n} d_i^2}{n(n^2-1)/2}, \tag{14}$$

where $N_{\text{con}}$ and $N_{\text{dis}}$ represent the number of concordant and discordant pairs in the two rankings, respectively. In Equation 14, $d_i$ represents the difference in rank for the $i$-th path in both rankings. For instance, if the true rank of $p_i$ is 1 and the predicted rank is 3, then $d_i = 1 - 3 = -2$.

## 5.3 Baselines

We compare with 8 path representation learning baselines. The details of methods are described as follows:

- **Node2vec** [10] is a graph embedding algorithm designed to learn continuous representations of nodes in a graph, effectively capturing both structural roles and community affiliations.
- **Toast** [4] is a road network representation framework that obtains effective road segment representations through traffic context aware skip-gram module and trajectory-enhanced transformer module.
- **PIM** [32] is an unsupervised path representation learning method that first employs curriculum learning to generate negative samples, followed by leveraging both global and local mutual information maximization to learn effective path representations.
- **Trembr** [9] is a representation learning framework based on a recurrent neural network, which incorporates the Road2Vec model to learn road segment embeddings.
- **PathRank** [36] is a supervised path representation learning method that employs Gated Recurrent Units (GRU) to model path sequences.
- **HMTRL** [22] utilizes a spatiotemporal graph neural network to capture autocorrelation and employs an attention module to generate route representations.
- **START** [17] is a self-supervised framework that utilizes a Graph Attention Network to convert road network features into segment embeddings.

**Table 1: Overall accuracy in travel time estimation and path ranking tasks. We use '↑' (and '↓') to indicate that larger (and smaller) values are better. For each task, we highlight the best and second-best performance in bold and underline. "Improvement" quantifies the enhancements achieved by Path-LLM over the best baseline.**

| Method | Xi'an | | | | | | Chengdu | | | | | |
|---|---|---|---|---|---|---|---|---|---|---|---|---|
| | Travel Time Estimation | | | Path Ranking | | | Travel Time Estimation | | | Path Ranking | | |
| | MAE ↓ | MARE ↓ | MAPE ↓ | MAE ↓ | $\tau$ ↑ | $\rho$ ↑ | MAE ↓ | MARE ↓ | MAPE ↓ | MAE ↓ | $\tau$ ↑ | $\rho$ ↑ |
| Node2vec | 5.926 | 0.300 | 34.801 | 0.175 | 0.551 | 0.614 | 4.539 | 0.309 | 38.367 | 0.28 | 0.519 | 0.571 |
| Toast | 5.786 | 0.295 | 33.742 | 0.180 | 0.629 | 0.672 | 6.151 | 0.424 | 61.585 | 0.320 | 0.394 | 0.415 |
| PIM | 4.512 | 0.228 | 24.892 | 0.143 | 0.664 | 0.737 | 3.450 | 0.241 | 27.934 | 0.192 | 0.570 | 0.628 |
| Trembr | 4.814 | 0.247 | 29.505 | 0.089 | 0.676 | 0.747 | 4.211 | 0.299 | 35.598 | 0.149 | 0.666 | 0.735 |
| PathRank | 4.696 | 0.234 | 25.735 | 0.155 | 0.667 | 0.735 | 3.610 | 0.255 | 30.854 | 0.231 | 0.618 | 0.674 |
| HMTRL | 4.516 | 0.228 | 25.786 | 0.164 | 0.676 | 0.749 | 3.557 | 0.246 | 28.897 | 0.259 | 0.653 | 0.721 |
| START | 4.642 | 0.234 | 26.313 | 0.116 | 0.682 | 0.753 | 3.328 | 0.229 | 26.237 | 0.231 | 0.671 | 0.739 |
| LightPath | 4.424 | 0.223 | 24.346 | 0.076 | 0.693 | 0.760 | 3.304 | 0.225 | 26.142 | 0.204 | 0.560 | 0.627 |
| **Path-LLM** | **4.179** | **0.207** | **20.885** | **0.029** | **0.766** | **0.819** | **3.096** | **0.213** | **23.290** | **0.093** | **0.706** | **0.772** |
| Improvement | 5.86% | 7.17% | 14.22% | 61.84% | 10.55% | 7.76% | 6.30% | 5.33% | 10.91% | 37.58% | 4.96% | 4.27% |

- **LightPath** [35] is a scalable and lightweight path representation learning framework based on sparse path encoder and relational reasoning.

## 5.4 Results and Analysis

*5.4.1 Overall Performance.* Table 1 presents the performance comparison of Path-LLM with all baseline models across two datasets under two downstream tasks. Overall, Path-LLM outperforms all the baselines in both datasets and across all evaluation metrics, demonstrating the superiority of our model. Specifically, we make the following observations.

Node2Vec performs poorly on downstream tasks across both datasets due to its limitations of graph embedding methods in capturing spatial relationships along a path. Both Toast and PathRank utilize bidirectional Gated Recurrent Unit (GRU) to capture sequential information. However, since Toast is primarily designed for trajectory representation learning, its architecture is more focused on modeling movement patterns. PIM relies on the construction of high-quality negative samples for their effectiveness, which results in the performance instability across different datasets and tasks. The performance of Trembr and HMTRL demonstrate the advantages of sequence modeling approaches in handling long sequence data. LightPath and START both employ contrastive learning approaches, and their performance surpasses that of most baseline models, indicating the significant advantage of contrastive learning in capturing distinctive features for path representation. However, their reliance solely on the road network topology modality may constrain their performance in complex paths. In contrast, Path-LLM integrates multi-modal information of paths, enabling a multifaceted understanding of paths that enhances the richness and accuracy of path representations, as well as improving performance across various tasks.

*5.4.2 Zero-shot Travel Time Estimation.* To evaluate the transferability of Path-LLM across different cities, we conduct the following zero-shot travel time estimation experiments: (1) Chengdu → Xi'an: the model is trained on the Chengdu dataset and evaluated

on the Xi'an dataset; (2) Xi'an → Chengdu: the model is trained on the Xi'an dataset and evaluated on the Chengdu dataset. These experiments aim to evaluate the generalization capability of Path-LLM on unseen datasets, particularly in cross-city scenarios where variations in the physical characteristics and regional features of paths may impact the model's performance. From the results in Table 2, it is evident that Path-LLM demonstrates significantly superior zero-shot performance compared to other baseline models.

*5.4.3 Few-shot Travel Time Estimation.* Evaluating the few-shot learning capability of Path-LLM under data-scarce conditions is also a key focus of our study. To simulate such scenarios, we conduct few-shot time estimation experiments on the Xi'an and Chengdu datasets, using 5% and 30% of the data samples, respectively, as shown in Table 3 and Table 4. The results show that Path-LLM exhibits robust generalization ability, even when the training data is extremely limited.

*5.4.4 Ablation Studies.* To evaluate the contribution of each component in Path-LLM, we conduct a comparative analysis between the full model and several variants:

- w/o Topology: This variant excludes the topological modality, relying only on the textual modality.
- w/o Text: This variant omits the textual modality, and uses only the topological modality as input.
- w/o TPfusion: This variant replaces the multimodal fusion module TPfusion with a linear layer.
- w/o TPalign: This variant removes the TPalign task, retaining only the TPfusion module.
- w/o TOCL: This variant randomly splits the dataset, and trains the model using randomly sampled batches.

We measure the performance of these variants across the Xi'an dataset on TTE and PR tasks, and the results are presented in Table 5. Based on the results, we make the following observations: (1) The variant w/o Topology significantly reduces performance, emphasizing the importance of topological information in both PR and TTE tasks; (2) The variant w/o Text also leads to performance

Table 2: Zero-shot learning in TTE task.

| Methods | Path-LLM | | | LightPath | | | START | | | PathRank | | | Trembr | | | HMTRL | | |
|---|---|---|---|---|---|---|---|---|---|---|---|---|---|---|---|---|---|---|
| Metric | MAE | MARE | MAPE | MAE | MARE | MAPE | MAE | MARE | MAPE | MAE | MARE | MAPE | MAE | MARE | MAPE | MAE | MARE | MAPE |
| Chengdu → Xi'an | **7.412** | **0.363** | **34.890** | 9.810 | 0.557 | 73.057 | 9.730 | 0.473 | 44.425 | 9.983 | 0.479 | 43.602 | 12.518 | 0.616 | 53.885 | 10.594 | 0.550 | 84.841 |
| Xi'an → Chengdu | **5.116** | **0.359** | **42.251** | 6.474 | 0.424 | 45.175 | 8.801 | 0.625 | 78.613 | 7.289 | 0.473 | 42.797 | 7.896 | 0.573 | 74.545 | 7.071 | 0.460 | 45.548 |

Table 3: Few-shot learning in TTE task on 5% label data.

| Methods | Path-LLM | | | LightPath | | | START | | | PathRank | | | Trembr | | | HMTRL | | |
|---|---|---|---|---|---|---|---|---|---|---|---|---|---|---|---|---|---|---|
| Metric | MAE | MARE | MAPE | MAE | MARE | MAPE | MAE | MARE | MAPE | MAE | MARE | MAPE | MAE | MARE | MAPE | MAE | MARE | MAPE |
| Chengdu | **3.649** | **0.251** | **29.705** | 3.728 | 0.259 | 30.793 | 4.170 | 0.291 | 36.331 | 4.662 | 0.304 | 30.692 | 4.983 | 0.357 | 43.500 | 4.607 | 0.299 | 28.983 |
| Xi'an | **4.481** | **0.222** | **23.265** | 4.807 | 0.241 | 26.955 | 5.515 | 0.278 | 31.651 | 7.160 | 0.381 | 49.656 | 6.164 | 0.294 | 29.746 | 7.357 | 0.395 | 52.901 |

Table 4: Few-shot learning in TTE task on 30% label data.

| Methods | Path-LLM | | | LightPath | | | START | | | PathRank | | | Trembr | | | HMTRL | | |
|---|---|---|---|---|---|---|---|---|---|---|---|---|---|---|---|---|---|---|
| Metric | MAE | MARE | MAPE | MAE | MARE | MAPE | MAE | MARE | MAPE | MAE | MARE | MAPE | MAE | MARE | MAPE | MAE | MARE | MAPE |
| Chengdu | **3.292** | **0.227** | **25.987** | 3.387 | 0.234 | 25.966 | 3.527 | 0.243 | 28.026 | 3.514 | 0.242 | 27.523 | 3.894 | 0.255 | 24.746 | 3.648 | 0.251 | 29.234 |
| Xi'an | **4.332** | **0.218** | **23.651** | 4.584 | 0.230 | 25.435 | 4.799 | 0.256 | 28.825 | 6.131 | 0.302 | 30.626 | 5.080 | 0.276 | 31.187 | 4.880 | 0.254 | 28.145 |

Table 5: Performance of Variants of Path-LLM

| Methods | Travel Time Estimation | | | Path Ranking | | |
|---|---|---|---|---|---|---|
| | MAE | MARE | MAPE | MAE | $\tau$ | $\rho$ |
| w/o Topology | 4.418 | 0.222 | 24.135 | 0.032 | 0.699 | 0.752 |
| w/o Text | 4.304 | 0.217 | 22.559 | 0.032 | 0.720 | 0.788 |
| w/o TPfusion | 4.519 | 0.226 | 24.449 | 0.030 | 0.716 | 0.787 |
| w/o TPalign | 4.226 | 0.212 | 22.025 | 0.032 | 0.729 | 0.793 |
| w/o TOCL | 4.238 | 0.213 | 22.756 | 0.030 | 0.723 | 0.789 |
| **Path-LLM** | **4.179** | **0.207** | **20.885** | **0.029** | **0.766** | **0.819** |

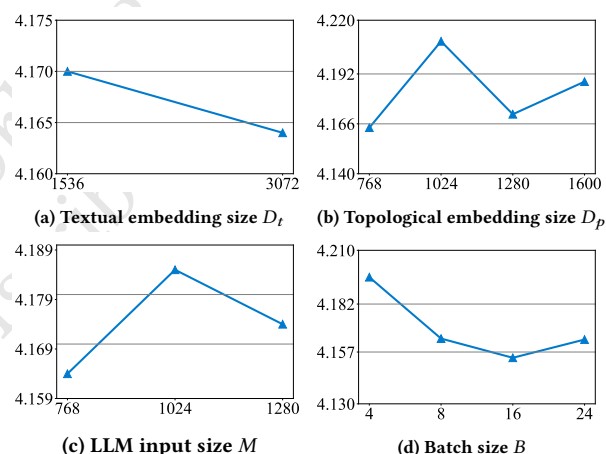

(a) Textual embedding size $D_t$  (b) Topological embedding size $D_p$

(c) LLM input size $M$  (d) Batch size $B$

Figure 5: Effects of hyper-parameters.

degradation, demonstrating the value of textual data in providing semantic insights and effectively leveraging pre-trained models for prior knowledge utilization; (3) The variant w/o TPfusion significantly reduces the performance of TTE predictions, highlighting the critical role of the TPfusion module in achieving effective multimodal data fusion. Additionally, The variant w/o TPalign results in performance declines, indicating that proper alignment between textual and topological embeddings is essential for maximizing the effectiveness of TPfusion in multimodal integration; (4) The variant w/o TOCL causes substantial degradation, showcasing the advantages of gradual learning for understanding multimodal path sequences.

*5.4.5 Impact of Hyper-parameters.* In this section, we explore how different hyper-parameters affect the performance of Path-LLM. Specifically, we evaluate the effects of textual embedding size $D_t$, topological embedding size $D_p$, LLM input embedding size $M$ and batch size $B$. Figure 5 presents the MAE results from the TTE task using Xi'an dataset. Similar trends are observed for the other tasks evaluated. The results indicate that increasing the text embedding size from 1536 to 3072 leads to a decline performance, and a topological embeddings size of $D_p$ achieves the best results. Increasing the LLM input embeddings results in performance degradation, likely due to the increased parameter count which makes fine-tuning more challenging. When the batch size $B$ is increased from 4 to 8, the MAE decreases significantly, but further increases offer only

marginal improvements. Therefore, setting $B = 8$ is the optimal choice when computational resources are limited.

## 6 CONCLUSIONS

In this work, we propose Path-LLM, a novel multi-modal path representation learning model that utilizes LLMs to integrate the topological, physical, regional, and functional characteristics of paths. We propose *TPalign*, a contrastive pretraining strategy to reduce modal conflicts by aligning multimodal embeddings. We also design *TPfusion* that dynamically adjusts modality importance. To train LLMs effectively, we introduce *TOCL*, a curriculum learning approach that gradually increases data complexity. Extensive experiments demonstrate the superior performance of Path-LLM. In the future, we aim to further explore multi-modal representation techniques for paths to achieve more comprehensive representations, which are critical for intelligent transportation systems.

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
