# OpenReview forum: "Path-LLM: A Multi-Modal Path Representation Learning by Aligning and Fusing with Large Language Models"
_ACM.org/TheWebConf/2025/Conference — WWW 2025 Poster_

### Official Review · Reviewer_48EW · 2024-11-30

**Novelty:** 7
**Technical Quality:** 7

**Review:**

This paper presents an innovative framework that introduces large language models (LLMs) for path representation learning, integrating topological, physical, and functional modalities. The authors design contrastive learning pretraining tasks at both the instance and feature levels to align modality differences and improve the performance of each modality. Additionally, the paper introduces the TOCL method, which progressively incorporates more challenging data to facilitate model training. The experimental results show that Path-LLM performs exceptionally well on two real-world datasets.

Overall, I think this paper offers the first novel solution to integrate LLMs into the path representation problem. My detailed comments are as follows:

### Strengths:

1. This paper is the first to introduce large language models (LLMs) for path representation learning, integrating topological, physical, and functional modalities. The proposed TPalign and TPfusion methods effectively integrate multimodal data, demonstrating strong innovation, particularly in addressing the alignment of topological and textual information.

2. This paper is well-structured and logically coherent, effectively conveying the motivations and challenges of incorporating multimodal data and LLMs.

3. Path-LLM performs exceptionally well on two real-world datasets across two downstreaem tasks, achieving up to a 61.84% improvement in path ranking performance on Xi’an dataset. It also demonstrates excellent performance in few-shot and zero-shot learning, providing high-quality path representations even in data-scarce situations, reducing the reliance on large amounts of labeled data, and offering significant practical value.

### Weaknesses:

1.	The explanation of the embedding distributions of the two modalities in Figure 1, as well as the discussion of the experimental details, could be further refined.

2.	The paper could further elaborate on the design of text prompts used to obtain text embeddings.

**Questions:**

1.	Despite the modal differences, please explain why the distribution of topological embeddings in Figure 1 is relatively compact, while the distribution of textual embeddings is more dispersed. Was this visualization based on real data?

2.	How is the prompt template for converting the road segment's name, type, and length into textual embeddings designed?

3.	Regarding the TOCL method proposed in the paper, please explain why the model only learns a subset of the training data in each round, yet this approach leads to improved performance. How does selecting a portion of the data in each iteration contribute to enhancing the model's learning and generalization?

4.	Is the λ in Equation 8 a learnable parameter? How is it set in the experiments?

**Reviewer Confidence:**

4: The reviewer is certain that the evaluation is correct and very familiar with the relevant literature

**Scope:**

4: The work is relevant to the Web and to the track, and is of broad interest to the community

---

### Official Review · Reviewer_HLbe · 2024-11-30

**Novelty:** 5
**Technical Quality:** 5

**Review:**

Path representation learning (PRL) is the fundamental task for location based applications and intelligent transportation systems. This paper argues that traditional PRL methods focus only on single-modality road network data, while ignoring physical and regional factors, which are also considered very important for traffic estimation.  With this in mind, this paper presents Path-LLM, a multimodal PRL that uses LLMs to facilitate the PRL task.  Experiments are conducted on a physical dataset and source codes are released.
Merits:
1. It is said that LLM  is firstly introduced to facilitate the PRL task.
2. Through contrastive learning and the two-stage overlapping curriculum learning, the proposed Path-LLM performs well across multiple tasks, outperforming many baselines.
3. The work has broad application prospects in intelligent transportation systems, mapping services, and etc.
4. The source codes are released.

Demerits:
1.Inconsistency & Lack of Explanation:
A) Inconsistency: the Abstract mentions that existing work has overlooked the importance of "physical and regional factors," while the Introduction states that physical (e.g., road length and width) and functional (e.g., road name and type) characteristics are important.
B) The Introduction introduces the term "road segment" without providing an explanation.
2. Challenges
The first and second Challenges are essentially related to the same thing: alignment or fusion of multi-modality data.Therefore, the Challenges facing PRL tasks still need to be refined.
3. Experimental Design:
A) Efficiency of the proposal should be evaluated.
B) The proposal should be compared with JGRM, which also adopts the idea of multiple modalities.
C) In Section 5.4.5 (Impact of Hyper-parameters): the authors only list conclusions but fail to provide the reasons for those conclusions.
4. Related works:
A) The paper lacks a thorough survey of related work, especially graphLLMs that also use topology and text modalities.
5. Other problems:
A) In Section 4.4.1, the meaning of x is unclear.
B) The significance of the red box in Figure 4 is explained, and the representation symbol for samples are changed from x to p.
C) Previous discussions of the paper focus on edge embeddings, but Figure 2 uses the term "Node Token".
D) Section 4.3 mentions that the proposed method uses the GPT-2 as the base model. WHY must be GPT-2? There are many excellent open-source language models have been proposed.

**Questions:**

1. Section 4.3 mentions that the proposed method uses the GPT-2 as the base model. WHY must be GPT-2? There are many excellent open-source language models have been proposed.
2. The first and second Challenges are essentially related to the same thing: alignment or fusion of multi-modality data.Therefore, the Challenges facing PRL tasks still need to be refined.
3. The paper lacks a thorough survey of related work, especially graphLLMs that also use topology and text modalities.

**Reviewer Confidence:**

4: The reviewer is certain that the evaluation is correct and very familiar with the relevant literature

**Scope:**

4: The work is relevant to the Web and to the track, and is of broad interest to the community

---

### Official Review · Reviewer_nMYp · 2024-12-02

**Novelty:** 4
**Technical Quality:** 3

**Review:**

This paper proposes a path representation method that incorporates multi-modal information and utilizes a large language model. Specifically, this article uses contrastive learning to strengthen the textual and topological embedding of paths from instance-level and feature-level, and leverages the gating mechanism to fuse them. Then, a large language model is fine-tuned in an easy-to-hard manner to obtain a unified path representation from the fused embedding.
Pros:
1.This paper is well-written, with clear and accessible overview of the related work, making it easy for readers to understand and follow.
2.The contrastive learning method for embedding alignment is explicitly displayed and explained, demonstrating its effect in integrating the interpretation and structural configuration of the multi-modal information.
3.Extensive ablation studies are conducted to verify the effectiveness of each proposed component in the framework.


Cons:
1.The technical contribution of proposed methods are not satisfactory enough. The gated fusion method in this article is a well-established approach in feature fusion~\cite{1, 2}, which linearly combining two embeddings with a weight learned from a neural network. Additionally, this paper proposes an easy-to-hard training strategy similar to that of~\cite{3}, and there was no evidence of model optimization when fine-tuning GPT-2. Therefore, the technical contribution of this article is relatively limited.
2.The rationale underpinning the proposed LLM lacks clarity. Neither topological embedding nor textual embedding is captured by the LLM, and the embedding fusion is completed prior to the introduction of the LLM, thereby rendering the necessity of the LLM in path representation learning.
3.The experimental framework outlined in this article is not persuasive enough, as it confines its analysis to a narrow range of downstream tasks and datasets, making it challenging to establish the robustness and generalization of the proposed method. Besides, the estimation of the LLM in this article is not sufficient enough, where experiments using other popular LLMs~(GPT-3, LLaMA, etc.) should be included as comparisons.

[1] Ke K, Li Z, Chen H, et al. Continuous Geodesic Self-Attention Models with Gated Fusion for Trajectory Prediction[C]//2024 IJCNN. IEEE, 2024: 1-7.
[2] Zhang Y, Zhong H, Chen G, et al. Multimodal Sentiment Analysis Network Based on Distributional Transformation and Gated Cross-Modal Fusion[C]//2024 NaNA. IEEE, 2024: 496-503.
[3] Yang S B, Guo C, Hu J, et al. Weakly-supervised temporal path representation learning with contrastive curriculum learning[C]//2022 ICDE. IEEE, 2022: 2873-2885.

**Questions:**

Q1.What is the functionality of the LLM if path embedding is aligned and fused before fed into LLM? You use large model methods to enhance the fused representation, rather than directly capturing language representations for further fusion?
Q2.The difficulty of the training data in this paper is assessed and ranked by SME. What is the physical meaning of the SME difficulty? Additionally, this paper cited another work that leverages expert models to estimate the difficulty of training data~\cite{3}, and this assessment sounds more rational, can you explain why you use MSE, instead of that one?

**Reviewer Confidence:**

2: The reviewer is willing to defend the evaluation, but it is likely that the reviewer did not understand parts of the paper

**Scope:**

3: The work is somewhat relevant to the Web and to the track, and is of narrow interest to a sub-community

---

### Official Review · Reviewer_5AAR · 2024-12-02

**Novelty:** 5
**Technical Quality:** 4

**Review:**

This paper proposes a multi-modal path representation learning model, Path-LLM, for path representation learning (PRL) in the transportation system. It integrates large language models (LLMs) into PRL to interpret both topological and textual data, enabling robust multi-modal path representations. To align and merge these modalities, it proposes TPalign, a contrastive learning-based pretraining strategy that ensures alignment within the embedding space. Then, it presents TPfusion, a multimodal fusion module that dynamically adjusts the weight of
each modality before integration. It also proposes Two-stage Overlapping Curriculum Learning (TOCL) to further optimize LLM training.  The experimental studies were conducted on two real datasets to show the effectiveness of the proposed method.

Pros:
S1: The paper proposes a new multi-modal path representation model that integrates large language models (LLMs) to capture topological and textual data.

S2. The proposed method outperforms the baselines for the  travel time estimation and path ranking tasks on two evaluated datasets.

 S3. The Figures, such as the architecture overview and the TOCL training process, can help readers understand proposed method.


Cons:
W1. The evaluated datasets are not publicly used in related works. The experiments are conducted on two datasets from cities Xi’an and Chengdu. It is unclear whether the proposed can also deal with and perform well on the datasets used in previous related works, such as the BJ and Porto datasets mentioned in [17] and the Aalborg, Denmark, and Synthetic datasets referenced in [35].

 W2. There is discussion or analysis on the computational cost of introducing LLMs into path representation learning. Moreover, aligning text and topological information by LLMs also has been studied in graph representation learning. Is there any specific challenges when aligning them in road networks?

 W3. The proposed method, which interprets both topological and textual data, seems similar to [17] that transforms road network features and travel semantics into representation vectors. There is no discussion/comparison to clarify their difference.

 W4. The traffic usually changes dynamically. It is unclear whether the proposed model can be deployed in the real systems to deal with the dynamically changing traffic data in a reasonable response time.

 W5.  The writing of the paper can be improved. For example:
(1)	In lines 56-57, page 1, P_1 and P_2 are wrongly defined, which is conflict with the data shown in Figure 1.
(2)	 In Figure 4, "Develop Curriculmn" contains a spelling mistake.
(3)	There is no labels on the x-axis and y-axis in Figure 5

**Questions:**

（1）Can the proposed method also work well on other datasets used in related work? For example, BJ and Porto datasets mentioned in [17] and the Aalborg, Denmark, and Synthetic datasets referenced in [35].

  (2) Can you clarify the specific difference between the proposed method and 17?

  (3) Can the proposed method efficiently deal with real-world road networks with dynamic changes to complete the tasks required by users in reasonable time?

**Reviewer Confidence:**

3: The reviewer is confident but not certain that the evaluation is correct

**Scope:**

4: The work is relevant to the Web and to the track, and is of broad interest to the community

---

### Official Review · Reviewer_H1DX · 2024-12-03

**Novelty:** 5
**Technical Quality:** 6

**Review:**

This paper presents Path-LLM, a multi-modal path representation learning framework integrating Large Language Models (LLMs) with textual and topological data. The authors propose TPalign for contrastive alignment and TPfusion for dynamic fusion, addressing modality conflicts. They also introduce a curriculum learning strategy (TOCL) to enhance generalization. Experimental results on real-world datasets demonstrate the model's superior performance in tasks like path ranking and travel time estimation, including in zero-shot and few-shot scenarios.

Strong points:
1、The integration of textual and topological modalities is well-designed, addressing key limitations of single-modal PRL methods.
2、The TOCL strategy effectively enhances generalization, especially in zero-shot and few-shot scenarios.
3、Empirical results demonstrate clear performance gains over strong baselines on real-world datasets.

Weak points:
W1. Although ablation studies are included, the paper does not deeply explore why certain components (e.g., TOCL or TPfusion) contribute significantly to performance, which could leave doubts about the necessity of these design choices.
W2. While the experiments are conducted on two datasets, the evaluation lacks diversity in terms of geographic and traffic conditions, which may not fully reflect the model’s adaptability to varied real-world scenarios.
W3. The reliance on pre-trained LLMs, while innovative, raises concerns about computational efficiency and scalability for real-time or large-scale transportation systems, especially given the need for fine-tuning and large memory footprints.

**Questions:**

O1. The paper evaluates Path-LLM on two datasets (Xi’an and Chengdu), but both are urban datasets with similar characteristics. Testing on datasets with different traffic patterns (e.g., rural or suburban areas) would improve the robustness of the evaluation.
O2. The evaluation focuses heavily on ranking metrics and Mean Absolute Error (MAE). However, metrics like F1 score or recall (for correct path prediction) could provide a more nuanced view of model performance, especially for downstream tasks like navigation.
O3. While the paper compares Path-LLM to eight baselines, some recent methods that incorporate multi-modal data (e.g., traffic signal information, real-time GPS updates) are missing. This limits the comprehensiveness of the benchmark comparison.
O4. The paper briefly mentions intelligent transportation systems but does not expand on how Path-LLM could specifically integrate with existing traffic management platforms or navigation services. Adding such discussions would enhance the paper’s practical impact.

**Reviewer Confidence:**

4: The reviewer is certain that the evaluation is correct and very familiar with the relevant literature

**Scope:**

3: The work is somewhat relevant to the Web and to the track, and is of narrow interest to a sub-community